# Towards General Error Diagnosis via Behavioral Testing in Machine Translation

**Junjie Wu[1], Lemao Liu[2], Dit-Yan Yeung[1]**

[1]The Hong Kong University of Science and Technology, Hong Kong SAR, China
[2]Tencent AI Lab
junjie.wu@connect.ust.hk, redmondliu@tencent.com, dyyeung@cse.ust.hk

## Abstract

Behavioral testing offers a crucial means of diagnosing linguistic errors and assessing capabilities of NLP models. However, applying behavioral testing to machine translation (MT) systems is challenging as it generally requires human efforts to craft references for evaluating the translation quality of such systems on newly generated test cases. Existing works in behavioral testing of MT systems circumvent this by evaluating translation quality without references, but this restricts diagnosis to specific types of errors, such as incorrect translation of single numeric or currency words. In order to diagnose general errors, this paper proposes a new **B**ilingual **T**ranslation **P**air **G**eneration based **B**ehavior **T**esting (**BTPGBT**) framework for conducting behavioral testing of MT systems. The core idea of BTPGBT is to employ a novel bilingual translation pair generation (BTPG) approach that automates the construction of high-quality test cases and their pseudo-references. Experimental results on various MT systems demonstrate that BTPGBT could provide comprehensive and accurate behavioral testing results for general error diagnosis, which further leads to several insightful findings. Our code and data are available at https://github.com/wujunjie1998/BTPGBT.

## 1 Introduction

In recent years, machine translation (MT) systems have achieved significant advancements in translation performance. Yet current MT systems are still brittle and could generate erroneous translations that lead to severe commercial losses [1], which makes error diagnosis of MT systems a crucial task to be solved. Behavioral testing, which has been widely applied in software engineering research to test complex systems, is a desired direction to tackle this issue. Generally, behavioral

---

[1]https://www.androidpolice.com/google-translation-mistake-io/

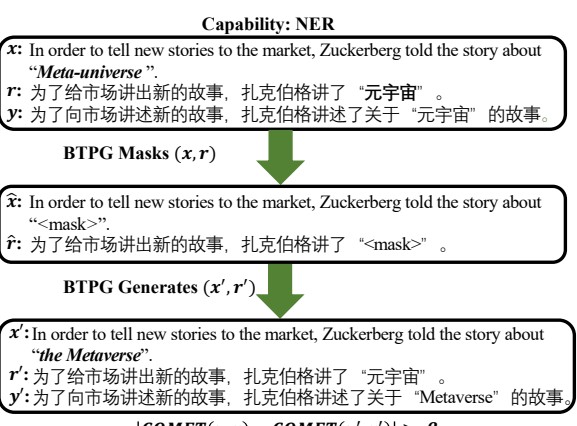

Figure 1: An example of behavioral testing the DeepL translator on the capability of translating named entities (NER). $x$ is the original input and $r$ is its reference. The masked counterparts of $x$ and $r$ are denoted as $\hat{x}$ and $\hat{r}$ respectively, with the to-be-masked segments highlighted in **boldface**. A generated test case, $x'$, replaces the named entity "*Meta-universe*" in $x$ with "*the Metaverse*", and its crafted pseudo-reference is denoted as $r'$. DeepL's outputs for $x$ and $x'$ are denoted as $y$ and $y'$. DeepL does not pass the behavioral testing on $x'$ since it outputs an erroneous $y'$ that directly copies "*Metaverse*" instead of translating it to "元宇宙", which is identified by the substantial difference in the COMET scores between $y$ and $y'$.

testing probes various system capabilities for error diagnosis by inspecting input-output behaviors on test cases with targeted edits, irrespective of knowledge about the system's internal structure (Beizer, 1995). Therefore, it is suitable for providing fine-grained evaluations of different capabilities and diagnosing errors for MT systems.

There are systematic studies of behavioral testing on classification tasks such as sentiment analysis (Ribeiro et al., 2020), hate speech detection (Röttger et al., 2021) and clinical outcome prediction (Van Aken et al., 2022). However, implementing behavioral testing in MT tasks poses significant challenges. The main reason is that it is usually difficult to automatically obtain the ref-

erence translation for a test case modified from a source sentence, making it difficult to automatically judge the quality of the translation of the test case. To avoid this challenge, existing work attempts to judge the translation quality of such test case without its reference translation, yet they can only diagnose translation errors related to specific types of edits that target certain capabilities. For example, He et al. (2020); Sun et al. (2020); Gupta et al. (2020); Ji et al. (2021) only diagnose translation errors on test cases with the editing of a single noun or adjective word, and He et al. (2021); Wang et al. (2021); Raunak et al. (2022) can only diagnose incorrect translation of noun phrases, quantities or currency units that is related to the edits on them.

To address these challenges, this paper presents **B**ilingual **T**ranslation **P**air **G**eneration based **Be**havior **T**esting (**BTPGBT**), a novel framework for behavioral testing in MT systems. The core idea of our framework is the novel Bilingual Translation Pair Generation (BTPG) approach that can automatically generate high-quality test cases and their pseudo-references from standard MT test sets, [2] which allows the diagnosis of general translation errors in test cases targeting various MT system capabilities. As shown in Figure 1, BTPG takes a source sentence and its reference as an input translation pair, then masks specific aligned segments ("*Meta-universe*" and "元宇宙" in Figure 1) in both sentences targeting the capability that needs to be tested (NER in Figure 1). Only masking aligned segments enables BTPG to make the best use of the structure information of unmasked positions in the original source and reference during generation, which enhances its generation quality. BTPG then generates a new bilingual translation pair as the test case and its pseudo-reference by using Chat-GPT [3], a large language model proposed by OpenAI that has shown impressive performances on various NLP tasks, to infill the masked segments in both sentences at once. In this way, the test case and its pseudo-reference could have similar quality to human constructed ones, as illustrated in §4.2.

With the high-quality test cases and their pseudo-references crafted by BTPG, we can conduct behav-

ioral testing to diagnose general translation errors targeting different capabilities. Extensive experiments on six MT systems using BTPGBT demonstrate that our method excels in evaluating multiple MT system capabilities with high error detection precision, outperforming prior works. Furthermore, our in-depth analysis of the testing results uncovers several insightful findings. The main contributions of this paper are summarized below:

1. We design a new framework that can automatically conduct general behavioral testing to diagnose MT systems. Specifically, we propose a novel bilingual translation pair generation method to generate high-quality test cases and corresponding pseudo-references for behavioral testing.

2. Through extensive experiments on six MT systems, we demonstrate our proposed method's effectiveness, leading to several insightful findings.

## 2 Revisiting Behavioral Testing in MT

### 2.1 Principled Definition

Behavioral testing (Beizer, 1995), which studies the capabilities of a system through examining its input-output behaviors on targeted test cases without knowing its internal information, has been demonstrated to be effective in NLP classification tasks (Ribeiro et al., 2020; Röttger et al., 2021; Van Aken et al., 2022). Following these works, the principled definition of behavioral testing in MT can be extended as: given an input sentence $x$ and its reference $r$. Let $x'$ be a test case edited from $x$ for testing a specific capability of an MT system $\mathcal{M}$ (e.g., $x'$ in Figure 1 targeting the NER capability), and $r'$ be the reference translation of $x'$. We say $\mathcal{M}$ passes the behavioral testing on $x'$ if it translates both $x$ and $x'$ in high-quality according to their references $r$ and $r'$. The idea behind this definition is straightforward: if $\mathcal{M}$ has the specific capability to handle the targeted edits in $x'$, it should translate $x'$ in the same quality as $x$.

Specifically, we first calculate the absolute difference of the quality between $y$ and $y'$, i.e., $\mathcal{M}$'s translations of $x$ and $x'$, as follows:

$$\mathrm{Diff}(\boldsymbol{y}, \boldsymbol{y'}) = |\mathrm{Qual}(\boldsymbol{y}) - \mathrm{Qual}(\boldsymbol{y'})| \quad (1)$$

Here we can apply various quality measuring strategies as Qual() to evaluate $y$ and $y'$, while it would

---

[2]Our framework indeed involves a test set with references, but this is not a very strong requirement since there are many off-the-shelf test datasets and in particular this is a standard setting in behavioral testing. For instance, the pioneered research about behavioral testing (Ribeiro et al., 2020) and many follow-up works (Röttger et al., 2021; Van Aken et al., 2022) all use a test dataset with ground-truth labels.

[3]https://chat.openai.com/

be more authentic if we use $r$ and $r'$ for evaluation. $\mathcal{M}$ passes the behavioral test on the test case $x'$ if it meets the following criteria:

$$\begin{cases} \mathrm{Qual}(y) \geq \alpha \\ \mathrm{Diff}(y, y') \leq \beta \end{cases} \quad (2)$$

where $\alpha$ and $\beta$ are thresholds ranging in $[0, 1]$. In practice, $\alpha$ should be high and $\beta$ should be low to ensure the testing effectiveness. Interpreting the above criteria is intuitive: if $\mathrm{Qual}(y)$ is higher than $\alpha$, we can conclude that $y$ is a high-quality translation of $x$. On this basis, if $\mathrm{Diff}(y, y')$ is lower than $\beta$, we can conclude that $y'$ keeps a similar and high translation quality with $y$ and thus $\mathcal{M}$ passes the behavioral testing. Otherwise, we say $\mathcal{M}$ does not pass the test and $y'$ is a diagnosed erroneous translation. As an example, the DeepL translator in Figure 1 fails on the behavioral testing on $x'$ since the erroneous translation $y'$ breaks Eq. 2, which is caused by the incorrect translation of "*Metaverse*" in $y'$. We use **Pass Rate**, i.e., the percentage of test cases that an MT system passes, to quantify the behavioral testing results, where a higher score means less translation errors.

## 2.2 Related Work: Challenges and Existing Solutions

However, it is challenging to craft a high-quality reference for each test case since it requires much human efforts. Hence, previous works propose various approaches without the references of test cases to diagnose translation errors on testing results.

**Existing Solutions for MT Behavior Testing.** (Wang et al., 2021; He et al., 2021; Raunak et al., 2022) construct a test case $x'$ by modifying special components like numbers, physical units and noun phrases in $x$, since the reference translations of these components can be obtained ahead from external dictionaries. They define $\mathcal{M}$ passes the behavioral testing on $x'$ if the reference translations of such components appear in $y'$. Although these behavioral testing methods hold a high precision in diagnosing translation errors, the errors found by them are limited to such specific components. In other words, the recall of translation errors identified by these approaches is low. On the other hand, (He et al., 2020; Sun et al., 2020; Gupta et al., 2020; Ji et al., 2021) first edit a single noun or adjective in $x$ to create a series of similar sentences as test cases, then denote $\mathcal{M}$ as passing

the behavioral testing if its translations of these sentences have similar syntactic structures, based on an assumption that the translations of similar sentences should be analogous. The reason why they only modify a single noun or adjective is to avoid largely shifting the structure of $x$, yet still limits the types of capability they can test as well as translation errors they can find. Further, even small modifications in $x$ could dramatically change its semantic meaning, making the assumption of these methods not stable and thus largely biases their corresponding evaluation results.

**Other Error Diagnosing Methods.** To provide fine-grained evaluation of MT systems, various evaluation methods except behavioral testing have been proposed, such as evaluating robustness to adversarial perturbations (Zhang et al., 2021; Lai et al., 2022), ambiguous words (Emelin et al., 2020) and diverse biases (Saunders and Byrne, 2020; Wang et al., 2022a). However, they all focus on a specific type of phenomenon in MT, thus lack the generality in diagnosing translation errors.

## 3 BTPGBT Framework

To solve issues in previous works, we propose a novel BTPGBT framework to conduct behavioral testing for general translation error diagnosis. Given a source sentence $x$ and its reference $r$, the core idea of BTPGBT is to automatically edit one or more segments in $x$ and $r$ to construct various test cases $x'$ and their pseudo-references $r'$ targeting specific capabilities of MT systems. Then we can diagnose these systems following the steps described in §2. In the following sections, we first describe how we implement the above core idea, then illustrate the capabilities BTPGBT tests as well as how to generate test cases targeting these capabilities for behavioral testing.

## 3.1 BTPG Method

However, implementing the core idea of BTPGBT is challenging due to two reasons. First, an appropriate strategy to decide which segment in $x$ and $r$ can be modified is needed, since we do not want the modification to largely shift the meaning of non-edited parts in $x$ and $r$ and hampers the generation quality. Second, how to edit the selected segments to craft fluent $x'$ and $r'$ is a non-trivial problem, while similar works like text infilling (Zhu et al., 2019; Donahue et al., 2020; Xiao et al., 2022) cannot be directly applied since we need to edit both $x$

and $r$ at once. To tackle these two issues and construct qualified $x'$ and $r'$ effectively, we present the bilingual translation pair generation (BTPG) approach and detail it in the following.

**Determining Modification Positions.** The first step of BTPG is to determine the segments that will be edited in the source sentence $x$, where segments here consist of words and phrases. Inspired by Knight (2000), we require that only a consecutive segment in $x$ that is solely aligned with another consecutive segment in $r$ can be edited, which avoids largely shifting the structures and meanings of non-edited segments. Take Figure 1 as an example. The segment "*In order to tell*" in $x$ cannot be edited since it is aligned with a non-consecutive segment "为了...讲出" in $y$. Specifically, we apply the Mask-Align tool proposed by Chen et al. (2021a) to obtain a word-to-word alignment between $x$ and $r$ for extracting two types of segments in $x$ that can be edited:

- a word in $x$ that only aligns with a word or a consecutive phrase in $r$.

- a consecutive phrase in $x$ that only aligns with a word or a consecutive phrase in $r$.

where phrases in $x$ and $r$ are identified by Stanza[4]. If a longer segment overlaps with another shorter segment, we will only keep the longer segment since it is usually more complete. Finally, we select one or several segments from the extracted segments for further editing.

**Generating Bilingual Translation Pairs.** Next, we illustrate how to craft a new bilingual translation pair $(x', r')$. Specifically, we first mask the selected segments in $x$ and its aligned segments in $r$ to construct a masked translation pair $(\hat{x}, \hat{r})$, then construct a new bilingual translation pair $(x', r')$ via filling in the masked positions in $(\hat{x}, \hat{r})$. However, existing text infilling approaches are conducted solely on source sentences (Zhu et al., 2019; Donahue et al., 2020) or target sentences (Chen et al., 2021b; Wang et al., 2022b; Xiao et al., 2022), and thus could not be adapted to our task since we need to fill in $\hat{x}$ and $\hat{r}$ at once.

Recently, the large language model ChatGPT designed by OpenAI has shown impressive performances on various NLP tasks including text generation and machine translation (Qin et al., 2023;

[4] https://stanfordnlp.github.io/stanza/

Jiao et al., 2023; Hendy et al., 2023; Peng et al., 2023), inspiring us to apply ChatGPT to generate $x'$ and $r'$. However, ChatGPT is still struggling with generating unbiased translations without hallucinations (Hendy et al., 2023; Guerreiro et al., 2023) when handling machine translation related tasks. To tackle this issue and make the best use of the power of ChatGPT, we propose to incorporate the masked translation pair $(\hat{x}, \hat{r})$ that contain semantic information of $(x, r)$ to build prompts for constructing a new bilingual translation pair. Given a masked pair $(\hat{x}, \hat{r})$, we instruct ChatGPT through its API to first fill in the masked position in $\hat{x}$, then fill in the masked positions in $\hat{r}$ based on the filled $\hat{x}$. We then guide ChatGPT to paraphrase the filled sentences to ensure fluency. After all, we obtained a new bilingual translation pair $(x', r')$ where $x'$ is the test case and $r'$ is its pseudo-reference. Through repeating the above steps, we can craft various test cases and their pseudo-references from $(x, r)$.

### 3.2 Constructing Test Cases Targeting Various Capabilities

With BTPG, we can make edits on any segments in $x$ that is aligned with another segment in $r$ to build test cases and their pseudo-references for behavioral testing. This generality enables BTPGBT to diagnose translation errors towards various capabilities of MT systems. In the following, we describe the nine types of MT capabilities we diagnose with BTPGBT as well as how to use BTPG to craft targeted test cases and their pseudo-references.

First, we consider a reliable MT system should appropriately understand the meanings and functions of segments that include words with different parts of speech. Therefore, we construct test cases with edits targeting seven types of parts of speech (POS). Next, it is crucial for MT systems to understand the tenses of different verbs in a sentence for correctly translating this sentence, thus we create test cases that edit the tense of a verb in a sentence to diagnose this capability (Tense). Finally, named entities that usually convey important information of a sentence should be translated properly. To this end, we build test cases with named entities-based edits to investigate MT systems' capability to properly translate named entities (NER). Except for modifying segments to other segments in the same category, we are also interested in MT systems' capability to handle uncertain modifications

| Capability | Description/Test Case Generation Instruction | Example |
|---|---|---|
| **POS** (Noun/Adjective/ Verb/Adverb/ Preposition/ Others) | Whether an MT system can understand the function and meaning of a segment in a source sentence that contains a word with certain POS tag. Test cases are created by replacing such segment in the source sentence with another segment that has a word with the same POS tag. | **Ori:** I printed out the page and took it to my local *shop*. **Test Case:** (Noun) I printed out the page and took it to my local *bookstore*. |
| **Tense** | Whether an MT system is able to identify and understand the tense of a verb in a source sentence and output a translation with the correct tense. Test cases are created by changing the tense of a non-past perfect tense verb in the source sentence to the past perfect tense. | **Ori:** Tracking number *will be provided* after dispatching the parcels. **Test Case:** Tracking number *had been issued* after dispatching the parcels. |
| **NER** | Whether an MT system can translate named entities (NEs) correctly. Test cases are crafted by changing a named entity in a source sentence to another named entity in the same NE category. | **Ori:** This Arab state plans to boost trade with *Russia* **Test Case:** This Arab state plans to boost trade with *Turkey*. |
| **General** | Whether an MT system can accurately understand and translate certain segments in a source sentence. Test cases are created by randomly modifying some segments in the source sentence, while limit the total number of words in the modified segments to be less than $0.2 \times \text{len}(\boldsymbol{x})$. | **Ori:** The stage when *Chinese companies* were *burning* money overseas on a large scale has passed. **Test Case:** The stage when *people* were *sending* money overseas on a large scale has passed. |

Table 1: Descriptions of different capabilities that BTPGBT evaluates for error diagnosis and instructions on how to edit corresponding test cases. Modified positions in each example sentence are in *italic*.

in the source sentence. Therefore, we craft test cases that do not limit the type and number of edits to examine this specific capability (General).

Detailed descriptions of each capability and the corresponding test cases generation instructions are shown in Table 1. For each capability, the selected segments that will be masked in $(\hat{\boldsymbol{x}}, \hat{\boldsymbol{r}})$ should also meet the condition shown in Table 1. Then we can craft test cases $\boldsymbol{x}'$ and their pseudo-reference $\boldsymbol{r}'$ with BTPG for behavioral testing.

## 3.3 Judging Behavioral Testing Results

With test cases and their pseudo-references in hand, we judge an MT system $\mathcal{M}$'s performance on a test case $\boldsymbol{x}'$ following the principled definition in §2.1 using its pseudo-reference $\boldsymbol{r}'$. To further enhance the reliability of Eq.2 when judging behavioral testing results, for each crafted test case $\boldsymbol{x}'$, we keep it only if $\text{Diff}(\boldsymbol{r}, \boldsymbol{r}') \leq \beta$, using the reference-free version of the widely used metric COMET (*wmt20-COMET-qe-da*) (Rei et al., 2022) as the quality measurement Qual(). This filtering step avoids the situation that $\text{Diff}(\boldsymbol{y}, \boldsymbol{y}') \leq \beta$ is caused by the difference between their corresponding references.

## 4 Experiments

We conduct experiments on the English-Chinese translation task. First, we implement both auto-

matic and human evaluations to measure the quality of the BTPG method. Next, we apply BTPGBT to diagnose six MT systems to show the effectiveness of our proposed behavioral testing framework, and perform in-depth analysis on the evaluation results to obtain some insightful findings.

### 4.1 Settings

We conduct our experiments using data extracted from the test sets of the WMT21/22 En-Zh/Zh-En news translation task [5] to build a dataset named **WMT** for our experiments, which are not included in the training data of our evaluated research MT systems. As for detailed settings of the BTPGBT framework, we apply the reference-based COMET-22 metric (*wmt22-COMET-da*) as the quality measurement Qual(). $\alpha$ and $\beta$ in Eq. 2 are set to 0.8 and 0.05, respectively [6].

### 4.2 Quality of BTPG

Since we aim at using BTPG to craft high-quality bilingual translation pairs as behavioral testing cases and their pseudo-references, we are interested in its generation quality. To this end, we randomly sample 200 source sentences and their

---

[5]https://www.statmt.org/wmt21/, https://www.statmt.org/wmt22/

[6]We provide details on how we set the value of the two hyperparameters in Appendix B

references from WMT and use BTPG to create 200 bilingual translation pairs as test cases and their pseudo-references for evaluation [7].

### 4.2.1 Evaluation Methods

We conduct human evaluations to evaluate the test cases and their pseudo-references generated by BTPG from three perspectives: source sentence fluency, target sentence fluency and translation adequacy. For translation adequacy, we also apply an automatic metric as an evaluation supplement.

**Source Sentence Fluency (SSF).** It measures whether the test case $x'$ is semantically meaningful and grammatically correct. For human evaluation, we ask human annotators to score the fluency of $x'$ based on a three-point rating scale, in which 1/2/3 represents unsatisfied/fair/satisfied, respectively.

**Target Sentence Fluency (TSF).** Similar to SSF, it measures whether the pseudo-reference $r'$ of $x'$ is semantically meaningful and grammatically correct. Likewise, we adopt the same three-point scale to score each $r'$ during human evaluation.

**Translation Adequacy (TA).** It captures whether the crafted pseudo-reference $r'$ of $x'$ has translated all the information conveyed by $x'$. For human evaluation, we ask annotators to compare $x'$ and $r'$ word-by-word and give a score for $r'$ following the same scoring scale as SSF. As for automatic evaluation, since we do not have a ground truth for $r'$, we apply the aforementioned *wmt20-COMET-qe-da* metric to measure translation adequacy [8], where a higher score indicates a more adequate translation.

For each human evaluation task, we invite three annotators who are proficient in both English and Chinese to rate the given text and they achieve a Krippendorff's alpha score (Krippendorff, 2011) of 0.82/0.80/0.80 on SSF/TSF/TA, respectively, indicating a high inter-agreement. See Appendix C for detailed instructions of our human evaluations.

### 4.2.2 Baselines

To illustrate the effectiveness of BTPG, we compare it with two baselines:

---

[7]To obtain more general evaluation results, we choose to target the General capability when generating test cases and skip the filtering step in §3.3.

[8]We do not use the latest reference-free *wmt22-COMETkiwi-da* metric since it was unavailable at the time of our experiments (May 2023).

|  | SSF | TSF | TA | COMET-20 |
|---|---|---|---|---|
| BART+BITIMT | 2.75 | 2.64 | 2.44 | 23.28 |
| Ref | **2.96** | **2.90** | 2.83 | 34.37 |
| BTPG | 2.93 | 2.83 | **2.85** | **36.38** |

Table 2: Quality evaluation results of test cases and their pseudo-references. SSF/TSF/TA are human annotated scores averaged among annotators.

• **BART+BiTIMT.** This approach is a combination of two text infilling models that generate $x'$ and $r'$ by first infilling the masked source $\hat{x}$, then infilling the masked reference $\hat{r}$. For source side infilling, we apply the widely used BART model (Lewis et al., 2020). As for the target side, we adopt the BiTIMT model (Xiao et al., 2022) to infill $\hat{r}$ on top of $x'$ filled by BART to craft $r'$.

• **Ref.** We are also interested in the quality of $x'$ and $r'$ crafted by BTPG compared to the source sentence $x$ and its reference $r$. Specifically, we use the 200 original translation pairs sampled from WMT as $x'$ and $r'$ for comparison.

### 4.2.3 Results

Table 2 lists the quality evaluation results. We observe that Ref and BTPG largely outperform BART+BiTIMT across the board, demonstrating the limitation of prior text infilling approaches that merely consider filling the masked positions without ensuring the fluency of the completed sentence. This characteristic also strictly lowers the translation performances of BART+BiTIMT, proven by its low TA and COMET scores. Conversely, $x'$ and $r'$ crafted by BTPG obtain comparable fluency scores with Ref, which ensures the quality of test cases and their pseudo-references constructed by BTPG. Surprisingly, BTPG achieves slightly higher TA and COMET scores than Ref, demonstrating that the pseudo-references $r'$ can act as the reference translation of crafted test cases $x'$. This conclusion is crucial since it supports our idea to conduct MT behavioral testing following the principled definition in §2 using the references of generated test cases.

### 4.3 Testing MT Systems with BTPGBT

Subsequently, we apply BTPGBT to examine various capabilities outlined in Table 1. For each capability except General (which is tested in Table 5), we randomly sample 1000 source sentences and their references from **WMT** and filter those that do not contain segments required by the tested ca-

| Task/Capability | Prompt Template |
|---|---|
| Direct Translation | ```"role": "system", "content": "You are a helpful assistant that translates English to Chinese.", "role": "user", "content": "[SRC]"``` |
| POS | ```"role": "system", "content": "You are a helpful assistant that translates English to Chinese.", "role": "user", "content": "You are given an English sentence and its Chinese translation. In each sentence, an {POS word/phrase} has been masked with the '<mask>' token. Your task is to first fill in the masked token in the English sentence using an {POS word/phrase} other than {the original POS word/phrase} without modifying any of the unmasked tokens. Then, use the filled English sentence to fill in the masked token in its corresponding Chinese translation. If necessary, make modifications to the filled Chinese translation to ensure fluency while preserving the meaning. Finally, please output the filled English sentence and its filled Chinese translation in the format of 'Filled English: \n Filled Chinese:'. \n English Sentence: [MASKED ENGLISH]. \n Chinese Translation: [MASKED CHINESE]"``` |

Table 3: Prompt templates for querying the ChatGPT model. Braces will be filled by modified segments in real usage.

pability (e.g., noun/noun phrase when testing the "Noun" capability). Then we use BTPGBT to construct test cases targeting these capabilities to test different MT systems and obtain the corresponding Pass Rates. For each capability in Table 1, we use a slightly different prompt for ChatGPT based on its instructions and list the prompt template targeting the POS capability in Table 3 as an example. The prompts targeting other capabilities are shown in Appendix §A. Noting that we only generate one test case for a given source sentence in this experiment, yet crafting more test cases can be simply done by repeating the generation process.

**MT Systems.** In this work, we evaluate various commercial translation systems, including Google Translate (**Google**) [9], Microsoft Azure Translate (**MS-Translator**) [10], DeepL Translate (**DeepL**) [11] and Tencent Transmart (**Tencent**) (Huang et al., 2021) [12]. We also evaluate a Transformer-based (Vaswani et al., 2017) NMT model as an evaluation supplement(**Transformer**) [13].

As mentioned in §3.1, ChatGPT has recently shown strong ability in machine translation tasks. Therefore, we also evaluate ChatGPT using BTPGBT. The prompt for instructing ChatGPT as a translator is shown in Table 3 (Direct Translation).

**Results.** Table 4 lists the behavioral testing results on different MT systems. We observe that all the other systems outperform Transformer on

all the capabilities, indicating the need for more advanced strategies in the design of robust MT systems. On the other hand, although ChatGPT largely outperforms Transformer, it still makes more translation errors compared to the top commercial MT system, particularly on test cases targeting Noun and Adv. This result illustrates that there still exists room for ChatGPT to become a stable translator. To our surprise, the commercial MT systems fail on 6%-17% of test cases targeting different capabilities, even when they are under regular testing and improvement. This finding further illustrates the importance of performing fine-grained error diagnosis with BTPGBT. Note that DeepL outperforms other commercial systems on all capabilities except Others, showing its strong ability to handle different types of edits in the input sentence.

**Large-scale Behavioral Testing.** Since BTPGBT can generalize large numbers of test cases and their pseudo-references efficiently, we also conduct behavioral testing on MT systems at scale to obtain unbiased error diagnosis. Concretely, we craft 20540 test cases targeting the General capability mentioned in Table 1 to test MT systems. As shown in Table 5, we notice that the performances of MT systems are similar to Table 4, where DeepL still outperforms other systems and all the systems outperform Transformer. These results further demonstrate that behavioral testing results in Table 4 are unbiased and could provide reliable error diagnosis of MT systems.

### 4.4 Effectiveness of BTPGBT

To further demonstrate the effectiveness of BTPGBT in diagnosing translation errors, we in-

---

[9] https://translate.google.com/
[10] https://learn.microsoft.com/en-us/azure/cognitive-services/translator/
[11] https://www.deepl.com/translator
[12] https://transmart.qq.com/zh-CN/index
[13] See Appendix D for details of this model.

| MT Systems | Noun | Verb | Adj | Adv | Prep | Others | NER | Tense |
|---|---|---|---|---|---|---|---|---|
| Google | 88.26 | 88.83 | 89.83 | 84.19 | 85.20 | **94.00** | 88.94 | 89.04 |
| MS-Translator | 87.25 | 83.25 | 91.49 | 85.58 | 86.23 | 91.07 | 87.00 | 89.37 |
| DeepL | **92.11** | **89.63** | **91.91** | **86.51** | **89.29** | 93.27 | **91.03** | **91.36** |
| Tencent | 88.87 | 84.84 | 88.80 | 85.58 | 88.27 | 91.07 | 88.34 | 87.04 |
| ChatGPT | 87.05 | 86.70 | 90.66 | 84.65 | 83.67 | 92.68 | 88.34 | 89.70 |
| Transformer | 70.24 | 69.68 | 71.99 | 68.84 | 69.39 | 79.80 | 71.30 | 70.76 |
| Avg | 85.63 | 83.82 | 87.45 | 82.56 | 83.68 | 90.32 | 85.83 | 86.21 |
| Size | 494 | 376 | 482 | 215 | 196 | 683 | 669 | 301 |

Table 4: Pass rates (%) of behavioral testing targeting eight different capabilities. Under each capability, the best results among all MT systems are **boldfaced**. Size refers to the number of test cases. Adj, Adv and Prep refers to the adjective, adverb and preposition capability in Table 1, respectively.

| MT System | Pass Rate |
|---|---|
| Google | 87.56 |
| MS-Translator | 86.30 |
| DeepL | **89.65** |
| Tencent | 86.02 |
| ChatGPT | 86.53 |
| Transformer | 69.36 |

Table 5: Large-scale behavioral testing results on 20540 test cases targeting the General capability.

| Method | Precision | Recall |
|---|---|---|
| **Dep** | 34.67 | 49.06 |
| **BTPGBT** | **82.61** | **71.70** |

Table 6: Precision (%) and recall (%) of Google's erroneous translations found by different methods.

$x$: Johnson is *teetering* on the edge of favour with his own MPs; if further photos surface, they may push him over the edge.
$y$: 约翰逊在他自己的国会议员的青睐边缘徘徊；如果进一步的照片浮出水面，可能会把他推到边缘。

$x'$: Johnson is *balancing* on the edge of favour with his own MPs; if further photos surface, they may push him over the edge.
$y'$: 约翰逊正在他自己的国会议员中取得平衡；如果进一步的照片出现，可能会把他推到边缘。
$y'$ Meaning: Johnson is balancing with his own MPs; if further photos surface, they may push him over the edge.

Figure 2: An example of behavioral testing of DeepL on the Verb capability. DeepL makes an error in $y'$ that it doe not translate "*on the edge of favour*" in $x'$.

that using pseudo-references to evaluate the quality of test case translations is beneficial for detecting more true translation errors.

### 4.5 Error Tendency

Notably, all MT systems in Table 4 tend to generate more translation errors towards the capabilities Verb and Adv, while achieving relatively high pass rates on Others. In this section, we study this phenomenon from the linguistic perspective. For the Others capability, we investigate the corresponding test cases and find that 98.60% of the test cases are related to numeric words substitution. Since numeric words usually do not affect the structures of sentences, it is not hard for MT systems to correctly translate different numeric words without affecting the translation of sentences, thus leading to high pass rates on test cases targeting the Others capability.

Conversely, a verb often forms collocations with different prepositions depending on the context. This factor makes it challenging for MT systems to correctly interpret the functions and meanings

vestigate its testing results from two perspectives: (1) the proportion of actual erroneous translations within the erroneous translations diagnosed by BTPGBT (**Precision**). (2) how many actual erroneous translations can BTPGBT identify (**Recall**). To this end, we first randomly select 100 distinct source sentences in WMT and pick their test cases crafted in the previous large-scale behavioral testing, then evaluate Google's translations on these test cases with BTPGBT. To calculate Precision and Recall scores, we manually annotate all the actual erroneous translations produced by Google on these test cases. Besides BTPGBT, we also experiment with the method designed by He et al. (2020) which identifies translation errors by comparing syntactic structures between original translations and translations of test cases (**Dep**), using its recommended configurations. As shown in Table 6, BTPGBT achieves much higher precision and recall scores compared to Dep, demonstrating

| MT Systems | Adj | Verb | NE |
|---|---|---|---|
| **Google** | 7.14 | **16.67** | **16.67** |
| **DeepL** | 5.88 | 15.79 | 14.82 |
| **ChatGPT** | **15.39** | 8.57 | 4.44 |

Table 7: Percentage (%) of erroneous translations that appear at the modified positions.

of verbs across various sentences, thus leading to errors when translating test cases with verb-based edits. An illustrative example is shown in Figure 2. Switching the verb "teetering" in $x$ to "balancing" should only change the sentence translation at the modified position. However, this edit makes the strongest MT system DeepL forget to translate "on the edge of favour" in $y'$, probably because DeepL pays much attention to the collocation "balance with" in $x'$. The translation difficulty of verbs may also affect MT systems' capability to translate adverbs and lead to the low pass rates on test cases targeting the Adv capability, since adverbs are often used to modify verbs in a sentence. In conclusion, through splitting evaluations of translation behaviors, researchers can systematically diagnose translation errors in MT systems, thereby facilitating more effective improvements.

### 4.6 Appearing Position of Errors

As demonstrated in Table 4, BTPGBT enables the identification of numerous translation errors in MT systems. Given that these errors arise from test cases with segment-level edits, there are two potential reasons for such translation errors:

- The segment-level edit directly induces the translation error, resulting in incorrect translation at the edited position, as exemplified in Figure 2.

- The segment-level edit indirectly induces the translation error. In this case, even though the edited position might be correctly translated, it affects the correct translation of other unedited parts of the test case and thus leads to translation errors.

To dive deeper into this phenomenon, we investigate whether the translation errors diagnosed by BTPGBT appear at the edited position, shedding light on why MT systems fail on such test cases. We annotate all the erroneous translations outputted by Google, DeepL and ChatGPT on three capabilities in Table 4, and calculate the percentage of

erroneous translations whose translation errors appear at the modified position. Results are shown in Table 7. We observe that few translation errors identified by BTPGBT appear at the modified position, indicating that existing MT systems are proficient at translating individual segments in a sentence, even when these segments are replaced. However, they lack the ability to comprehend changes in a sentence's structure caused by segment modifications, which consequently leads to translation errors in non-modified positions.

## 5   Conclusion

In this paper, we introduce a novel behavioral testing framework BTPGBT to diagnose general translation errors for MT systems. The key idea of BTPGBT is to auto-generate test cases and their pseudo-references, which facilitates the diagnosis of translation errors targeting various capabilities of MT systems. To this end, we design the BTPG approach to craft test cases and pseudo-references, whose effectiveness has been proven by extensive experiments. Experimental results on six different MT systems further demonstrate that BTPGBT can provide general and faithful behavioral testing results and lead to some insightful findings.

## Limitations

In our proposed behavioral testing framework BTPGBT, we incorporate a ChatGPT-equipped module to automatically craft test cases and their pseudo-references for general error diagnosis. However, ChatGPT does not keep the same translation and text generation quality on all languages, which may hinder us applying BTPGBT on low-resource translation tasks. This characteristic requires us to introduce additional methods under these scenarios. We will study this direction in our future work.

## Ethical Considerations

Since the processes of crafting pseudo-references is totally automatic, some toxic and harmful test cases might be generated when querying the ChatGPT model. It is also possible that some toxic and harmful translations are outputted by commercial MT systems when they make translation errors on the test cases. These potential issues require the actual users to perform comprehensive data post-processing before conducting behavioral testing

using BTPGBT, and make sure that the found translation errors will not be used under non-debugging situations.

## Acknowledgement

This research has been made possible by funding support from the Research Grants Council of Hong Kong under the General Research Fund project 16204720 and the Research Impact Fund project R6003-21.

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

# A  Additional Prompts for ChatGPT

In this section, we list the prompt templates used for querying ChatGPT to craft test cases targeting the **Tense** and **NER** capability in Table 8 for reference as a supplement to Table 3.

# B  Details on Setting Hyperparameter Values

In §4.1, we mentioned that we set the values of the two hyperparameters $\alpha$ and $\beta$ as 0.8 and 0.05, respectively. In this section, we provide more details about how these two values are selected.

Unlike behavioral testing on NLP classification tasks where errors can be directly derived from the prediction outputs, we cannot directly identify testing errors from MT outputs. Therefore, we propose $\alpha$ and $\beta$ to help identify translation errors from the MT outputs automatically. As shown in Equation 2:

- $\alpha$ controls a user's degree of acceptance of MT systems' outputs. If $\text{Qual}(\boldsymbol{y}) \geq \alpha$, we say that the user regards $\boldsymbol{y}$ as a good translation and accepts it for further testing. In the paper, $\alpha$ is set to 0.8 since we use COMET as the quality evaluation metric and we regard $\boldsymbol{y}$ as a good translation if $\text{Qual}(\boldsymbol{y}) \geq 0.8$.

- $\beta$ controls a user's tolerance on translation errors and if $\text{Diff}(\boldsymbol{y}, \boldsymbol{y}') \leq \beta$, we say that the quality of $\boldsymbol{y}'$ (the MT system's output on the modified test case) is similar to $\boldsymbol{y}$ and the MT system passes the behavioral test. In this paper, $\beta$ is set to 0.05 since we believe that a COMET difference lower than 0.05 is a reasonable shift caused by the modifications in the test cases. If $\text{Diff}(\boldsymbol{y}, \boldsymbol{y}') \leq 0.05$, we regard $\boldsymbol{y}$ and $\boldsymbol{y}'$ have similar translation quality and the MT system passes the behavioral test.

Moreover, we would like to illustrate that the value setting of the two hyperparameters will not affect the conclusions drawn from the evaluation results. To demonstrate this point, we perform an experiment by 1): varying the value of $\alpha$ while fixing $\beta$; 2): varying the value of $\beta$ while fixing $\alpha$, and obtaining the corresponding pass rates for the

six MT systems on test cases targeting the Noun, Verb and NER capabilities. Specifically, we set 1): $\alpha$ as 0.5/0.6/0.7/0.8 while fixing $\beta = 0.05$; 2): $\beta$ as 0.02/0.05/0.08/0.11 while fixing $\alpha = 0.8$, and list the results in Table 9, Table 10 and Table 11, respectively.

From the three tables, we observe that the values of the two hyperparameters will not affect the conclusions obtained from Table 4:

- Transformer obtains the worst results.

- In most cases, ChatGPT makes more translation errors compared to the top commercial MT system.

- DeepL outperforms other commercial MT systems.

- MT systems make more errors on test cases targeting the Verb capability.

In conclusion, we point out the following points regarding the hyperparameter value setting:

- $\alpha$ is set to 0.8 since we use COMET as the quality evaluation metric and we regard $\boldsymbol{y}$ as a good translation if $\text{Qual}(\boldsymbol{y}) \geq 0.8$. $\beta$ is set to 0.05 since we believe that a COMET difference lower than 0.05 is a reasonable shift caused by the modifications in the test cases.

- The values of these two hyperparameters will not affect the conclusions obtained from the experiments.

# C  Detailed Human Evaluation Instructions

The detailed instructions of our human evaluation tasks are shown in Table 12.

# D  Details of the Transformer-based MT System

In this section, we provide details of the Transformer-based MT system introduced in §4.3, which contains six 512-dimensional encoder and decoder layers, respectively. The word-embedding size is set to 512. Two separate BPE models are applied to generate two vocabularies for English ($\sim$48K tokens) and Chinese ($\sim$59K tokens). We train Transformer using the WMT20 En-Zh news translation corpus that includes 31M sentence pairs (Bojar et al., 2017) for 300K steps with 6K warmup steps using the cosine learning rate

| Task/Capability | Prompt Template |
|---|---|
| Tense | "role": "system", "content": "You are a helpful assistant that translates English to Chinese.", "role": "user", "content": "You are given an English sentence and its Chinese translation. In each sentence, a verb/verb phrase has been masked with the '<mask>' token. Your task is to first fill in the masked token in the English sentence using a past perfect tense verb/verb phrase without modifying any of the unmasked tokens. Then, use the filled English sentence to fill in the masked token in its corresponding Chinese translation in the past perfect tense. If necessary, make modifications to the filled Chinese translation to ensure the correctness of tense while preserving the meaning. Finally, please output the filled English sentence and its filled Chinese translation in the format of 'Filled English: \n Filled Chinese:'. \n English Sentence: [MASKED ENGLISH]. \n Chinese Translation: [MASKED CHINESE]" |
| NER | "role": "system", "content": "You are a helpful assistant that translates English to Chinese.", "role": "user", "content": "You are given an English sentence and its Chinese translation. In each sentence, an {named entity type} has been masked with the '<mask>' token. Your task is to first fill in the masked token in the English sentence using an {named entity type} other than {the original named entity} without modifying any of the unmasked tokens. Then, use the filled English sentence to fill in the masked token in its corresponding Chinese translation. If necessary, make modifications to the filled Chinese translation to ensure fluency while preserving the meaning. Finally, please output the filled English sentence and its filled Chinese translation in the format of 'Filled English: \n Filled Chinese:'. \n English Sentence: [MASKED ENGLISH]. \n Chinese Translation: [MASKED CHINESE]" |

Table 8: Prompt templates for querying the ChatGPT model. For the NER capability, we use Stanza to identify named entities in a sentence and the "named entity type" refers to one of the 18 named entity types provided by Stanza.

| MT System | $\alpha = 0.5,$ $\beta = 0.05$ | $\alpha = 0.6,$ $\beta = 0.05$ | $\alpha = 0.7,$ $\beta = 0.05$ | $\alpha = 0.8,$ $\beta = 0.02$ | $\alpha = 0.8,$ $\beta = 0.08$ | $\alpha = 0.8,$ $\beta = 0.11$ | $\alpha = 0.8,$ $\beta = 0.05$ (Table 4) |
|---|---|---|---|---|---|---|---|
| Google | 94.74 | 94.33 | 93.31 | 73.08 | 91.09 | 91.30 | 88.26 |
| MS-Translator | 94.94 | 94.74 | 93.52 | 73.48 | 89.27 | 89.27 | 87.25 |
| DeepL | **95.95** | **95.95** | **95.14** | **79.15** | **93.18** | **93.18** | **92.11** |
| Tencent | 94.53 | 93.93 | 91.90 | 72.67 | 90.49 | 91.09 | 88.87 |
| ChatGPT | 90.89 | 90.89 | 89.87 | 64.58 | 90.69 | 90.89 | 87.05 |
| Transformer | 85.83 | 84.21 | 80.16 | 54.45 | 72.47 | 73.68 | 70.24 |
| Avg | 92.81 | 92.34 | 90.65 | 69.57 | 87.86 | 88.24 | 85.63 |

Table 9: Pass rates (%) of behavioral testing targeting the Noun capability under different hyperparameter settings. Under each setting, the best results among all MT systems are **boldfaced**. The last column refers to the corresponding results listed in Table 4.

scheduling strategy. During training, we use the Adam optimizer ($\beta_1 = 0.9, \beta_2 = 0.98$) with a learning rate of 1e-4. The dropout rate is set to 0.3 and the label smoothing rate is set to 0.2. The maximum number of tokens in a batch is set to 65536. Transformer achieves an average BLEU score of 38.66 on the WMT20 En-Zh test set [14].

---

[14]For reference, the rank 1st system (Shi et al., 2020) on this set achieves an average BLEU score of 43.20.

| MT System | $\alpha = 0.5,$ $\beta = 0.05$ | $\alpha = 0.6,$ $\beta = 0.05$ | $\alpha = 0.7,$ $\beta = 0.05$ | $\alpha = 0.8,$ $\beta = 0.02$ | $\alpha = 0.8,$ $\beta = 0.08$ | $\alpha = 0.8,$ $\beta = 0.11$ | $\alpha = 0.8,$ $\beta = 0.05$ (Table 4) |
|---|---|---|---|---|---|---|---|
| Google | 93.62 | 93.35 | 92.02 | 67.02 | **92.02** | 92.55 | 88.83 |
| MS-Translator | 89.89 | 89.89 | 88.30 | 66.46 | 88.56 | 90.96 | 83.25 |
| DeepL | **94.95** | **94.95** | **94.42** | **69.68** | 91.76 | 92.29 | **89.63** |
| Tencent | 90.96 | 90.43 | 89.36 | 67.82 | 89.10 | 89.89 | 84.84 |
| ChatGPT | 90.69 | 90.69 | 90.16 | 60.64 | **92.02** | 92.82 | 86.70 |
| Transformer | 85.37 | 82.71 | 79.26 | 51.33 | 71.54 | 73.94 | 69.68 |
| Avg | 90.91 | 90.34 | 88.92 | 63.83 | 87.50 | 88.74 | 83.82 |

Table 10: Pass rates (%) of behavioral testing targeting the Verb capability under different hyperparameter settings. Under each setting, the best results among all MT systems are **boldfaced**. The last column refers to the corresponding results listed in Table 4.

| MT System | $\alpha = 0.5,$ $\beta = 0.05$ | $\alpha = 0.6,$ $\beta = 0.05$ | $\alpha = 0.7,$ $\beta = 0.05$ | $\alpha = 0.8,$ $\beta = 0.02$ | $\alpha = 0.8,$ $\beta = 0.08$ | $\alpha = 0.8,$ $\beta = 0.11$ | $\alpha = 0.8,$ $\beta = 0.05$ (Table 4) |
|---|---|---|---|---|---|---|---|
| Google | 95.52 | 95.37 | 95.07 | 76.98 | 91.33 | 92.08 | 88.94 |
| MS-Translator | 95.07 | 94.77 | 93.57 | 75.19 | 89.39 | 89.99 | 87.00 |
| DeepL | **95.96** | **95.96** | **95.67** | **77.73** | **93.57** | **93.87** | **91.03** |
| Tencent | 94.62 | 94.62 | 93.57 | 76.68 | 90.28 | 91.03 | 88.34 |
| ChatGPT | 93.27 | 93.27 | 92.98 | 63.68 | 92.38 | 93.27 | 88.34 |
| Transformer | 87.44 | 87.00 | 83.11 | 58.45 | 73.54 | 74.29 | 71.30 |
| Avg | 93.65 | 93.50 | 92.33 | 71.45 | 88.42 | 89.09 | 85.83 |

Table 11: Pass rates (%) of behavioral testing targeting the NER capability under different hyperparameter settings. Under each setting, the best results among all MT systems are **boldfaced**. The last column refers to the corresponding results listed in Table 4.

**Source sentence fluency**: 1/2/3. For each source sentence, you need to determine which description category it belongs to and mark it with the corresponding score. Your need not to consider the translation, just consider the source sentence itself.

| Score | Description |
| --- | --- |
| 1 (unsatisfied) | 1) The text is totally broken, or contains severe grammar errors. 
 or 
 2) The text is very hard to understand. |
| 2 (fair) | 1) The text is basically fluent, but contains few grammar errors that do not affect understanding. 
 or 
 2) The text is basically fluent, but contains some repeated context. 
 or 
 3) The text is basically fluent, but contains perverse or fake content that is obviously different from the commonsense. |
| 3 (satisfied) | 1) The text is fluent and do not have grammatical errors/repeated context/perverse or fake content. It is not hard to understand the meaning of the sentence. |

**Translation Fluency**: 1/2/3. For each translation, you need to determine which description category it belongs to and mark it with the corresponding score. You need not to consider the source sentence, just consider the translation it self.

| Score | Description |
| --- | --- |
| 1 (unsatisfied) | 1) The text is totally broken, or contains severe grammar errors. 
 or 
 2) The text is very hard to understand. |
| 2 (fair) | 1) The text is basically fluent, but contains few grammar errors that do not affect understanding. 
 or 
 2) The text is basically fluent, but contains some repeated context. 
 or 
 3) The text is basically fluent, but contains perverse or fake content that is obviously different from the commonsense. |
| 3 (satisfied) | 1) The text is fluent and do not have grammartical errors/repeated context/perverse or fake content. It is not hard to understand the meaning of the sentence. |

**Translation Adequacy**: 1/2/3. For each translation, you need to determine which description category it belongs to and mark it with the corresponding score. Please consider the source sentence and the translation together. You need not to consider the fluency of the translation if there is no difficulty in understanding.

| Score | Description |
| --- | --- |
| 1 (unsatisfied) | 1) The translation is irrelevant to the source sentence. 
 or 
 2) Many parts of the source sentence are missed in the translation. |
| 2 (fair) | 1) Few parts of the source sentence are missed in the translation. 
 or 
 2) All parts of the source sentence have been translated, but the tenses/syntactic structures of the source sentence and the translation are different. 
 or 
 3) Some parts that are not in the source sentences have been translated. |
| 3 (satisfied) | 1) All the parts of the source sentence are translated correctly. The tenses and the syntactic structures of both the source sentence and the translation are also the same. |

Table 12: Detailed human evaluation instructions.