# OpenReview forum: "Towards General Error Diagnosis via Behavioral Testing in Machine Translation"
_EMNLP/2023/Conference — EMNLP 2023 Findings_

### Official Review · Reviewer_Qgqn · 2023-07-29

**Soundness:** 4

**Excitement:**

4: Strong: This paper deepens the understanding of some phenomenon or lowers the barriers to an existing research direction.

**Paper Topic And Main Contributions:**

This paper proposes and evaluates a method to generate block box test cases for machine translation (MT) systems. Given an off-the-shelve test set (the authors use the WMT21/22 news translation test sets), test cases are generated by modifying specific features of segments in the test set by, for example, replacing a named entity in both the source segment and its reference translation. The replacement is based on a number of open-source tools (Mask-Align, Stanza) as well as a closed chat-optimised LLM (ChatGPT). An MT system passes a test case if an automatic MT quality evaluation metric (COMET in this paper) (1) assigns a ‘good’ score to the system's translation of the original source segment and (2) the difference between the score for the translation of the original source segment and the modified source segment is ‘small’. ‘good’ and ‘small’ are defined as hyperparameters alpha and beta in the paper.

**Questions For The Authors:**

* On what basis did you choose 0.8 and 0.05 for the hyperparameters alpha and beta in your experiments? This is crucial since the pass/fail percentages for the systems tested are heavily influenced by these parameters. I am willing to increase my score if the authors can convincingly describe how the parameters were chosen and how they could be optimised if someone were to reproduce this work (e.g., with other languages or MT evaluation metrics).
* When did you obtain the ChatGPT outputs? Assuming you used it via OpenAI's API, which model did you use exactly?

**Reasons To Accept:**

* The authors convincingly show that their method enables the generation of valid test cases through human evaluation in terms of source sentence fluency, pseudo-reference fluency, and translation adequacy between the two.
* The proposed method scales well. While previous work in automatic test case construction has focussed primarily on simple modifications (such as numbers), the method proposed here can, in theory, create an infinite number of test cases by varying any linguistic feature.

**Reasons To Reject:**

* The authors experiment with a single language direction (English to Chinese). Since ChatGPT is a seminal component of the proposed pipeline, it is to be expected that the approach will work with other high-resource, but not (or not as good) with low(er)-resource language pairs.
* The hyperparameters alpha and beta are crucial since they determine whether or not a constructed test case fails. The authors present the percentage of successful test cases for six commercial MT systems, but these numbers all depend on the aforementioned hyperparameters. The paper contains no indication of why the values of 0.8 and 0.05 are used and how they were derived/optimised.
* The proposed method relies heavily on ChatGPT, which makes the results presented in the paper hard to reproduce. (The method presented in the paper as such, however, could be easily reproduced.)

**Reproducibility:**

1: Could not reproduce the results here no matter how hard they tried.

**Reviewer Confidence:**

4: Quite sure. I tried to check the important points carefully. It's unlikely, though conceivable, that I missed something that should affect my ratings.

**Typos Grammar Style And Presentation Improvements:**

* Remove bold formatting of ‘2’ in Table 2.
* L457: ‘Noting’ should be ‘Note’.

---

> ### Author Rebuttal · Authors · 2023-08-29
>
> We thank you for your constructive comments and feedback! We address your concerns as follows:
>
> ### Reasons to reject
>
> > ChatGPT is to be expected to work with other high-resource, but not (or not as good) with low(er)-resource language pairs.
>
> While we agree that ChatGPT's direct translation results on low-resource languages may not be as good as those on high-resource languages and have incorporated this point in the limitation section, recent works have proposed several ways to remedy this issue. For example, Lu et al. (2023) [1] propose to incorporate the knowledge of external multilingual dictionaries to design prompts for ChatGPT on low-resource translation task, which largely enhances ChatGPT's performances on such tasks. In future works, we're considering integrating such advancements into our methodology to enhance its capability for low-resource language translation tasks.
>
> > On what basis did you choose 0.8 and 0.05 for the hyperparameters alpha and beta in your experiments? And how they could be optimised if someone were to reproduce this work (e.g., with other languages or MT evaluation metrics)
>
> First of all, we'd like to emphasize that the exact values of these two hyperparameters are derived from the user's tolerance level towards translation errors. Therefore, we choose values for $\alpha$ and $\beta$ based on our tolerance towards translation errors without tuning. We will detail this point in the following.
>
> We then explain the meaning of these two hyperparameters (Line 149-157) and why we select 0.8 and 0.05 for them. Unlike behavioral testing on NLP classification tasks where errors can be directly derived from the prediction outputs, we cannot directly identify testing errors from MT outputs. Therefore, we propose $\alpha$ and $\beta$ to help identify translation errors from the MT outputs automatically.
>
> As shown in Equation 2,
>
> * $\alpha$ controls an user's degree of acceptance of MT systems' outputs. If $Qual(y) \geq \alpha$, we say that the user regard $y$ as a good translation and accept it for further testing. **In the paper, $\alpha$ is set to 0.8 since we use COMET as the quality evaluation metric and we regard $y$ as a good translation if $Qual (y) \geq 0.8$**.
>
> * $\beta$ controls an user's tolerance on translation errors and if $Diff(y, y') \leq \beta$, we say that the quality of $y'$ (the MT system's output on the modified test case $x'$) is similar to $y$ and the MT system passes the behavioral test. **In this paper, $\beta$ is set to 0.05 since we believe that an COMET difference lower than 0.05 is a reasonable shift caused by the modifications in the test cases**. If $Diff(y, y') \leq \beta$, we regard $y$ and $y'$ have similar translation qualities and the MT system passes the behavioral test.
>
> Moreover, we'd like to illustrate that the value setting of the two hyperparameters will not affect the conclusions draw from the evaluation results, and perform an additional experiment to prove this point. Specifically, we vary the value of $\alpha$ (while fixing $\beta$) / vary the value of $\beta$ (while fixing $\alpha$) and obtain the corresponding pass rates for the six MT systems on test cases targeting the **Noun**, **Verb** and **NER** capabilities. We set $\alpha$ as 0.5/0.6/0.7/0.8/ (while fixing $\beta=0.05$) / $\beta$ as 0.02/0.05/0.08/0.11 (while fixing $\alpha=0.8$) and list the results as follows (values in the last column are the results in Table 3):
>
>  **Noun**
> |   MT System     | $\alpha=0.5,\beta=0.05$ | $\alpha=0.6,\beta=0.05$ | $\alpha=0.7,\beta=0.05$ |$\alpha=0.8,\beta=0.02$ | $\alpha=0.8,\beta=0.08$ | $\alpha=0.8,\beta=0.11$ | $\alpha=0.8,\beta=0.05$ (**our setting**) |
> |--------|  :----------:  |  :----------: |  :----------: |  :----------: |  :----------: |  :----------: |  :----------: |
> | Google | 94.74    | 94.33    | 93.31    |  73.08    | 91.09    | 91.30    | 88.26    |
> | MS-Translator     | 94.94    | 94.74    | 93.52    |  73.48    | 89.27    | 89.27    | 87.25    |
> | DeepL  | **95.95**    | **95.95**    | **95.14**    | **79.15**    | **93.18**    | **93.18**    | **92.11**    |
> | Tencent| 94.53    | 93.93    | 91.90    |  72.67    | 90.49    | 91.09    | 88.87    |
> | ChatGPT   | 90.89    | 90.89    | 89.87    | 64.58    | 90.69    | 90.89    | 87.05    |
> | Transformer  | 85.83    | 84.21    | 80.16    |  54.45    | 72.47    | 73.68    | 70.24    |
>
>  **Verb**
> |   MT System     | $\alpha=0.5, \beta=0.05$ | $\alpha=0.6, \beta=0.05$ | $\alpha=0.7, \beta=0.05$ | $\alpha=0.8, \beta=0.02$ | $\alpha=0.8, \beta=0.08$ | $\alpha=0.8, \beta=0.11$ | $\alpha=0.8, \beta=0.05$ (**our setting**)  |
> |--------|--------------------------|--------------------------|--------------------------|--------------------------|--------------------------|--------------------------|--------------------------|
> | Google | 93.62                     | 93.35                    | 92.02                    | 67.02                    | **92.02**                    | 92.55                    | 88.83                    |
> | MS-Translator     | 89.89                     | 89.89                    | 88.30                    | 66.46                    | 88.56                    | 90.96                    | 83.25                    |
> | DeepL  | **94.95**                     | **94.95**                    | **94.42**                    | **69.68**                    | 91.76                    | 92.29                    | **89.63**                    |
> | Tencent| 90.96                     | 90.43                    | 89.36                    | 67.82                    | 89.10                    | 89.89                    | 84.84                    |
> | ChatGPT   | 90.69                     | 90.69                    | 90.16                    | 60.64                    | **92.02**                    | **92.82**                    | 86.70                    |
> | Transformer  | 85.37                     | 82.71                    | 79.26                    | 51.33                    | 71.54                    | 73.94                    | 69.68                    |
>
>  **NER**
>
> |        | $\alpha=0.5, \beta=0.05$ | $\alpha=0.6, \beta=0.05$ | $\alpha=0.7, \beta=0.05$ | $\alpha=0.8, \beta=0.02$ | $\alpha=0.8, \beta=0.08$ | $\alpha=0.8, \beta=0.11$ | $\alpha=0.8, \beta=0.05$ (**our setting**) |
> |--------|--------------------------|--------------------------|--------------------------|--------------------------|--------------------------|--------------------------|--------------------------|
> | Google | 95.52                     | 95.37                    | 95.07                    | 76.98                    | 91.33                    | 92.08                    | 88.94                    |
> | MS-Translator     | 95.07                     | 94.77                    | 93.57                    | 75.19                    | 89.39                    | 89.99                    | 87.00                    |
> | DeepL  | **95.96**                     | **95.96**                    | **95.67**                    | **77.73**                    | **93.57**                    | **93.87**                    | **91.03**                    |
> | Tencent| 94.62                     | 94.62                    | 93.57                    | 76.68                    | 90.28                    | 91.03                    | 88.34                    |
> | ChatGPT   | 93.27                     | 93.27                    | 92.98                    | 63.68                    | 92.38                    | 93.27                    | 88.34                    |
> | Transformer  | 87.44                     | 87.00                    | 83.11                    | 58.45                    | 73.54                    | 74.29                    | 71.30                    |
>
> From the above results, we observe that the values of the two hyperparameters will not affect the conclusions in our paper:
> 1. Transformer obtains the worst results.
> 2. In most cases, ChatGPT makes more translation errors compared to the top commercial MT system.
> 3. DeepL outperforms other commercial MT systems.
> 4. MT systems make more errors on test cases targeting the **Verb** capability.
>
> **In conclusion, we point out that**:
>
> 1. The exact values of these two hyperparameters are selected based on the degree of users' tolerance towards translation errors.
>
> 2. $\alpha$ is set to 0.8 since we use COMET as the quality evaluation metric and we regard $y$ as a good translation if $Qual (y) \geq 0.8$. $\beta$ is set to 0.05 since we believe that an COMET difference lower than 0.05 is a reasonable shift caused by the modifications in the test cases. We do not tune these two hyperparameters in our work.
>
> 3. The values of these two hyperparameters will not affect the conclusions in our paper.
>
> **When reproducing our method with a new metric or on a new task, we recommend users to**:
>
> 1. Know what value of this metric represents good translation on a certain task, and what value represents large difference between translation outputs.
>
> 2. Understand your own degree of tolerance towards translation errors, and transfer it into hyperparameter values.
>
> 3. Make sure that different values of $\alpha$ and $\beta$ will not affect the conclusions obtained from the evaluation results.
>
> > The proposed method relies heavily on ChatGPT, which makes the results presented in the paper hard to reproduce.
>
> We promise to open-source all the codes, all the generated test cases, all the translation outputs generated by the six MT systems for exactly reproducing our experiments. When reproducing our work with other translation tasks, our proposed method can be easily adapted and thanks for your appreciation on this point.
>
> ### Questions for the authors
>
> ### Q1:
> Please refer to our second parts of reply above.
>
> ### Q2:
>
> We obtained ChatGPT 's outputs in April and May, 2023. The model we used was Open AI's "gpt-3.5-turbo" (it seems that it is now called "gpt-3.5-turbo-0613").
>
> ### Typos
>
> Thanks for the suggestions. We will revise these typos in the revised version.
>
>
> ### References in the rebuttal
>
> [1] Lu, H., Huang, H., Zhang, D., Yang, H., Lam, W., & Wei, F. (2023). Chain-of-Dictionary Prompting Elicits Translation in Large Language Models. arXiv preprint arXiv:2305.06575.

---

### Official Review · Reviewer_59vL · 2023-08-01

**Soundness:** 3

**Excitement:**

3: Ambivalent: It has merits (e.g., it reports state-of-the-art results, the idea is nice), but there are key weaknesses (e.g., it describes incremental work), and it can significantly benefit from another round of revision. However, I won't object to accepting it if my co-reviewers champion it.

**Paper Topic And Main Contributions:**

The paper presents a method to automatically create behavioral tests for MT from an existing parallel corpus. The method involves word-aligning the parallel sentences, selecting parallel phrases (similarly to what was done in SMT, but selecting only syntactic phrases) and replacing them, both in source and target, with likely segments in that context by means of chatGPT. A test is passed if the automatic evaluation metric score is similar to that of the original parallel sentence, as long as the original parallel sentences obtained a good enough score. A human evaluation showed that the generated sentence pairs are adequate and fluent.

The main contributions of this work are the following:
- The algorithm itself, that can be used to create tests for MT systems and automatically detect specific errors in MT systems, going beyond the usual automatic evaluation metrics
- The validation, by means of human evaluation, of the capabilities of large language models (chatGPT) for mask infilling in a multilingual way

**Questions For The Authors:**

Question A: When determining modification positions, as you always choose the longest segment when there are overlaps, doesn’t it prevent you from selecting single-word segments?
Question B: How exactly do you generate multiple test cases from the same parallel sentence? What happens if there is only one possible phrase pair that can be masked?

**Reasons To Accept:**

The presented algorithm can help to easily detect errors in MT systems without additional parallel data

**Reasons To Reject:**

The proposed algorithm is difficult to reproduce. Neither in section 3.1 nor in the rest of the paper, it is clearly stated how to proceed when more than one non-overlapping segment can be extracted from a sentence. The ChatGPT prompts, that are one of the main components of the algorithm, are not shown in the main paper. In section 3.3, it is not clear how Diff(r, r’) is computed, since later it is stated that reference-based COMET is used as the evaluation metric. It is also not clear how the number of test cases generated from the same source sentence can be increased (it is mentioned in Section 4.3), as no mention is made about how the prompts should be changed. Regarding the experiments on Large-scale behavioral testing, there are not enough details about how the 20540 tests were obtained. It is not clear which were the original sentences and how many segments were extracted from each parallel sentence.

The analyses carried out using the proposed algorithm are not really informative when it comes to drawing conclusions about MT systems (Sec. 4.3). They represent how often a sentence built by modifying a phrase that contains a word with a particular POS is translated “correctly”. Firstly, the error may come from another word from the phrase and, secondly, according to Table 6, the errors usually happen in parts of the target sentence that have not been modified during the creation of the test case.

Concerning technical soundness, chatGPT should not be used as an MT system being evaluated since it has been used to generate the test cases, thus its translation quality could be overestimated.

**Reproducibility:**

3: Could reproduce the results with some difficulty. The settings of parameters are underspecified or subjectively determined; the training/evaluation data are not widely available.

**Reviewer Confidence:**

2: Willing to defend my evaluation, but it is fairly likely that I missed some details, didn't understand some central points, or can't be sure about the novelty of the work.

**Typos Grammar Style And Presentation Improvements:**

Please avoid including information that is crucial to understand the paper in Appendixes, such as the chatGPT prompts, related works and the training data the Transformer system is trained on.

---

> ### Author Rebuttal · Authors · 2023-08-29
>
> We thank you for your constructive comments and feedback! We address your concerns as follows:
>
> ### Reasons to reject
>
> > The proposed algorithm is difficult to reproduce.
>
> We promise to make all our code, generated test cases, and translation outputs from the six MT systems publicly available to ensure complete reproducibility of our experiments. For the issues regarding reproducing the algorithm, we will answer them in the following.
>
> > Neither in section 3.1 nor in the rest of the paper, it is clearly stated how to proceed when more than one non-overlapping segment can be extracted from a sentence.
>
> We have provided the relevant details in the "Test Case Generation Instruction" column of Table 1 (Lines 328-330) . For the **POS**, **Tense**, and **NER** capabilities, only a segment containing a word with a POS tag/verb/named entity in the source sentence will be modified, irrespective of how many segments are derived from this sentence. However, for the **General** capability, if there are multiple segments extracted from the source sentence, more than one of them can be modified.
>
> > The ChatGPT prompts, that are one of the main components of the algorithm, are not shown in the main paper.
>
> We put all the prompt templates in the appendix (Table 7) and will put them in the main paper in the revised version.
>
> > In section 3.3, it is not clear how Diff(r, r’) is computed
>
> Since the proposed BTPG method enables us to follow the principle definition to conduct behavioral testing (Section 2.1), we use equation (1) to calculate $Diff(r,r')$. For this calculation, we employ the reference-free variant of COMET to determine $Qual(r)$ and $Qual(r')$ (refer to Line 336-344 and 397-402). We will make the distinction clearer in the revised version.
>
> > ... how the number of test cases generated from the same source sentence can be increased (it is mentioned in Section 4.3), as no mention is made about how the prompts should be changed.
>
> When generating multiple test cases from the same source sentence, the change of the prompt is only on the target segment that will be modified (mentioned in Table 7). Since we can extract multiple candidate segments from the same source sentence, we just need to use different segment as the target segment and repeat the generation process to obtain multiple test cases, as described in line 256-257, line 293-295 and line 457-460.
>
> > Regarding the experiments on Large-scale behavioral testing, there are not enough details about how the 20540 tests were obtained, which were the original sentences and how many segments were extracted from each parallel sentence.
>
> We use the dataset *WMT* to generate the 20540 test cases targeting the **General** capability (Line 355-360). As described in Line 243-257, all the aligned segments that fulfill our requirements are extracted from each parallel sentence.
>
> We also provide more statistics of the data and the segments. Overall, there are 6862 source sentences and their references in *WMT*, where 41240 segments can be extracted in total (6.01 for each parallel sentence in average). Moreover, considering the combinations between segments,we can construct 28484862025 masked pairs ($\hat{x}$, $\hat{r}$) targeting the **General** capability for further test case generation. ChatGPT's diverse blank-filling behavior also allows us to craft a vast number of unique test cases with ease.
>
> >  The analyses carried out using the proposed algorithm are not really informative since the translation errors may not happens at the modified position.
>
> First, we want to again emphasize the contribution of this paper, i.e, **we conduct behavior testing in machine translation, which reveals more translation error cases for well-known MT systems (including the Google translate) with less human efforts**.
>
> To this end, we first propose a novel method that can automatically generate large numbers of test cases from some off-the-shelve MT test sets (also mentioned by Reviewer Qgqn). Then we use this method to conduct behavior testing for six well-known MT systems. Compared to previous fine-grained error analysis works in MT that requires much human efforts on crafting high-quality references, our method alleviate this issue with much cheaper costs.
>
> Regarding the latter contribution (Section 4.3, 4.4, 4.5), the "pass rate" in Section 4.3 is crucial for diagnosing MT systems since it directly tells us what and how many errors MT systems may encounter on a batch of test cases targeting various capabilities, then lead to insightful findings (Section 4.3, Section 4.5) just like previous NLP behavioral testing works have done (Ribeiro et al., 2020) [1]. Specifically, two main types of translation errors are identified in our experiments:
>
> **Type 1**.  The modified segment is translated incorrectly.
>
> **Type 2**.  The modified segment is translated correctly. However, it makes the MT system incorrectly translates another segment and not passing the behavioral test.
>
> To dive deeper into the above sentence-level translation errors, in Section 4.5, we investigate how these two kinds of errors distribute in all the founded error cases, i.e., whether a translation error occurs at the modified position (Type 1) or at other positions (Type 2) and draw several conclusions in Section 4.4 and Section 4.5 and obtain some insightful findings from our behavioral testing results.
>
> > ChatGPT should not be evaluated since its translation quality could be over-estimated.
>
> First, we'd like to clarify that **the focus of this paper is to evaluate existing MT systems, including commercial MT systems and strong open-sourced MT systems**. Since these MT systems are not incorporated in the test case generation process, their evaluation results are not biased.
>
> For ChatGPT, we admit that its evaluation results might be over-estimated. However, as shown in Table 3, ChatGPT's pass rates are significantly lower than the best commercial MT system, which is in line with the conclusions in existing works that ChatGPT performs worse than the strongest commercial MT systems [2] [3]. Such observations affirm that the over-estimate concern linked to ChatGPT isn't substantial, further showing the robustness of our proposed method and we put ChatGPT's results in our paper to make our evaluation more comprehensive.
>
> ### Questions for the authors
>
> ### Question A
> Given that a sentence typically comprises more single words than phrases, numerous single-word segments remain in the candidate list, even when we consistently opt for the longest segment in cases of overlap. For reference, of the 41240 segments derived from *WMT*, 35302 are single-word segments, while 5938 are phrasal segments.
>
> ### Question B
>
> In our experiments, if there is only one possible segment that can be masked in the source sentence, we only generate one test case with this source sentence. However, given the diversity of ChatGPT's outputs, we can easily generate multiple test cases by querying ChatGPT for multiple times, while making sure that there is no overlap in the crafted test cases.
>
> When a source sentence has multiple segments, as mentioned above, we use different segment as the target segment and modify the prompt of ChatGPT correspondingly, then repeat the generation process to obtain multiple test cases (line 256-257, line 293-295 and line 457-460).
>
> ### Typos
> Thanks for your suggestion. We will revise these issues in the revised version.
>
> ### References in the rebuttal
>
> [1] Ribeiro, M. T., Wu, T., Guestrin, C., & Singh, S. (2020). Beyond accuracy: Behavioral testing of NLP models with CheckList. arXiv preprint arXiv:2005.04118.
>
> [2] Jiao, W., Wang, W., Huang, J. T., Wang, X., & Tu, Z. (2023). Is ChatGPT a good translator? A preliminary study. arXiv preprint arXiv:2301.08745.
>
> [3] Hendy, A., Abdelrehim, M., Sharaf, A., Raunak, V., Gabr, M., Matsushita, H., ... & Awadalla, H. H. (2023). How good are gpt models at machine translation? a comprehensive evaluation. arXiv preprint arXiv:2302.09210.

---

### Official Review · Reviewer_5gEN · 2023-08-04

**Soundness:** 3

**Excitement:**

3: Ambivalent: It has merits (e.g., it reports state-of-the-art results, the idea is nice), but there are key weaknesses (e.g., it describes incremental work), and it can significantly benefit from another round of revision. However, I won't object to accepting it if my co-reviewers champion it.

**Paper Topic And Main Contributions:**

This paper presents a method for creating a test set for behavioral testing of machine translation systems.
The test set is produced by first masking a pair of corresponding words/phrases in a translation pair and then filling them by using ChatGPT.
Quality evaluation of the produced test set revealed that the rewritten translation pairs are as good as their original translation pairs.
Six MT systems are compared using the automatically generated test set.

**Questions For The Authors:**

* Please provide the number of the translation errors for which the precision and the recall in Table 5 was calculated.

**Reasons To Accept:**

The motivation is clear.
The generated test set is shown to be of good quality.

**Reasons To Reject:**

The quality test of the generated test set is only done for the category “general” while, as far as I understand, one of the advantage of the behavioral test is that it can test a certain aspect of the systems’ capability by comparing the result across different test set categories. The author explains the reason in footnote 8 but I could not get the meaning of “more general evaluation result.” Since there is no guarantee that the quality of the “general” category is at the same level of the other, more targeted, categories, it appears to be a weak point of the paper.

Another concern I had about the paper is that it is not very clear what we can get from the behavioral test using the generated test cases. The author does not draw much information from the evaluation results of the six MT systems (Section 4.3). It is suggested that the relatively low “pass rate” on verb category is due to the fact that the accurate translation of a verb often requires to take into account the collocation with prepositions. It is however just exemplified with a single instance and we do not know how many of the test failures can be explained by it. Furthermore, in Section 4.5, the author says most of the translation failure identified by their method is **not** on the modified position of the test cases (e.g., the replaced verbs in the “verb” test cases). Isn’t it contradict with the interpretation of the “verb” category result in Section 4.3?

**Reproducibility:**

3: Could reproduce the results with some difficulty. The settings of parameters are underspecified or subjectively determined; the training/evaluation data are not widely available.

**Reviewer Confidence:**

3: Pretty sure, but there's a chance I missed something. Although I have a good feel for this area in general, I did not carefully check the paper's details, e.g., the math, experimental design, or novelty.

**Typos Grammar Style And Presentation Improvements:**

L210: follow the steps -> follow**ing** the steps ?

L255: between x and r -> ??

L406: (?)  [bibtex entry is missing?]

Table 3, header: Adp -> Adj ?

---

> ### Author Rebuttal · Authors · 2023-08-29
>
> We thank you for your constructive comments and feedback! We address your concerns as follows:
>
> ### Reasons to reject:
>
> > The quality test of the generated test set is only done for the category “general”
>
> We first want to clarify the relationship of the **General** capability and other capabilities. Our motivation of evaluating the **General** capability is to test MT systems under a more complex and random scenario. Different from test cases targeting other capabilities that only edit one specific type of segment in the source sentence, test cases targeting the **General** capability may edit one or several random segments (mentioned in its "Description/Test Case Generation Instruction" column in Table 1) simultaneously and thus leads to more general and diverse testing examples.
>
> Next, based on the definition of the **General** capability, two kinds of test cases could be constructed:
>
> 1. Test cases entailing modification to just one segment. These can also serve as test cases for other distinct capabilities such as **POS**. Importantly, during our experiments, we ensured that there was no overlap between test cases for the **General** capability and other capabilities.
>
> 2. Test cases involving modifications to several segments at once, which are not suitable as test cases for the other specific capabilities.
>
> In addition, we pick out test cases that fulfill the requirement of the **POS** capability (modifying one segment with a certain POS tag)  from the 200 tested pairs used in Section 4.2, and calculate their quality scores separately as follows:
>
>  POS (24 test cases)
> |     |  **SSF** | **TSF** | **TA** | **COMET-20** |
> |  ----  | ----  | ---- | ---- | ---- |
> |  BART+BITIMT | 2.83 | 2.86 | 2.64 | 32.42 |
> |  Ref |**2.96**  |**2.95** |2.80 |41.23 |
> |  BTPG |**2.96**  |2.93|**2.86** |**41.36** |
>
> We can see that the quality evaluation results above are similar to Table 2. Namely, BTPG showcases results in line with Ref on "SSF" and "TSF" while surpassing both Ref and BART+BITIMT in "TA" and "COMET-20" evaluations. Therefore, we conclude that test cases targeting the **General** capability are comprehensive and representative and are suitable for the quality evaluation (Section 4.2).
>
> In conclusion, we conduct quality evaluation on test cases targeting the **General** capability since such test cases are more comprehensive and representative. Moreover, we show that results obtained from test cases targeting the **General** capability are similar to results targeting specific capabilities like **POS**, which further proves our quality evaluation strategy. We will make this point clearer in the revised version.
>
> > ... it is not very clear what we can get from the behavioral test using the generated test cases.
>
> First, we want to again emphasize the contribution of this paper, i.e, we conduct behavior testing in MT, which reveals more translation error cases for well-known MT systems (including the Google translate) with less human efforts. To this end, we first propose a novel method that can automatically generate large numbers of test cases from some off-the-shelve MT test sets (also mentioned by Reviewer Qgqn). Then we use this method to conduct behavior testing for six well-known MT systems. Compared to previous fine-grained error analysis works in MT that requires much human efforts on crafting high-quality references, our method alleviate this issue with much cheaper costs.
>
> Regarding the latter contribution, the "pass rate" of behavioral testing in Table 3 is crucial for diagnosing MT systems since it directly tells us how many errors MT systems may encounter on a batch of test cases. And these founded errors can lead to some insightful findings (Section 4.3, Section 4.5) just like previous NLP behavioral testing works have done (Ribeiro et al., 2020) [1].
>
> > Analysis of low error rates on verbs is however just exemplified with a single instance and we do not know how many of the test failures can be explained by it.
>
> We manually examine all the errors generated by MS-translator and Google targeting the **Verb** capability in Table 3. The percentages of translation errors caused by verbs' collocations with different prepositions are 60.32% and 40.48%, respectively. This result illustrates that this error type does dominate the translation errors targeting the **Verb** capability.
>
> > Section 4.5 contradicts with the interpretation of the “verb” category result in Section 4.3.
>
> In Section 4.3, we focus on sentence-level translation errors caused by verb modification regardless of the error position, which can be directly identified by the "pass rate" as shown in Table 3. Within these identified discrepancies, we can pinpoint two specific translation error types associated with the collocation of the modified verb with prepositions:
>
> Type 1. The modified verb is translated incorrectly due to its collocation with prepositions in the test case, making the MT system not passing the behavioral test.
>
> Type 2. The modified verb is translated correctly. However, it collocates with a preposition that is originally collocated with another segment $s$ in the test case, thus making the MT system incorrectly translates $s$ and not passing the behavioral test.
>
> Taking the case in Figure 2 as an example. The modified verb *“balancing”* in the test case $x'$ makes DeepL forget to translate *“on the edge of favour”* in $y'$, probably because DeepL pays more attention to the collocation *“balance with”* in $x'$ than the original collocation *"on the edge of favour with"* (Line 512-518).
>
> To dive deeper into the above sentence-level translation errors, in Section 4.5, we investigate how these two kinds of errors distribute in all the founded error cases. Specifically, we question whether an error is predominantly positioned at the modified site (Type 1) or elsewhere (Type 2). This analysis is not contradicting with the analysis in Section 4.3. We will make this point clearer in the revised version.
>
> ### Questions for the authors
>
> When calculating erroneous translations in Table 5 (Line 548-555), we experiment on 135 test cases constructed from the 100 source sentences.
>
> The number of translation errors founded by Dep and BTPGBT are 75 and 46, respectively. Among these errors, 26 and 38 are regarded as true by humans, respectively. Therefore, the precision values of Dep and BTPGBT are 26/75 $\approx$ 34.67% and 38/48 $\approx$ 82.61%, respectively.
>
> Next, according to human evaluation, the Google translation system makes 53 actual translation errors on all the test cases. Therefore, the recall values of Dep and BTPGBT are 26/53 $\approx$ 49.06% and 38/53 $\approx$ 71.70%, respectively.
>
> > Typos
>
> Thanks for the suggestions. We will revise these typos in the revised version.
>
> ### References in the rebuttal
>
> [1] Ribeiro, M. T., Wu, T., Guestrin, C., & Singh, S. (2020). Beyond accuracy: Behavioral testing of NLP models with CheckList. arXiv preprint arXiv:2005.04118.

---

### Meta-Review · Area_Chair_215S · 2023-09-17

**Recommendation:** 2

**Metareview:**

The paper proposes a behavioural evaluation framework for MT. The method first masks a pair of aligned words or phrases in the source and the reference translation and fills them in with ChatGPT. The evaluation test: the model passes the test if (i) the original translation quality is good, (ii) the translation quality of the new pair is similar to the old quality. For English and Chinese, the authors show that the new sentences, i.e. the ones used for evaluation, are of good quality. The authors test six MT systems.

The reviewers feel rather ambivalent about the paper. While they acknowledged that the test set could help finding MT errors without additional parallel data, they all had the same concerns. Firstly, it is not clear what to take out of the evaluation results: (i) often the models do not pass the tests because of an error outside of the modified span, (ii) many words/phrases cannot be translated in isolation (e.g., verbs often require collocations with prepositions). Secondly, the core of the method is using ChatGPT to translate: while this might work for the single language pair considered in the paper (English-Chinese), it is largely not applicable for other languages. While the authors replied very extensively and were extremely proactive during discussion, I believe these concerns are (i) valid and (ii) not addressed (e.g., mentioning the use of ChatGPT in the limitation section does not revoke the concern).

---

### Decision · Program_Chairs · 2023-10-07

**Decision:**

Accept-Findings

**Comment:**

The paper proposes a behavioural evaluation framework for MT. The method first masks a pair of aligned words or phrases in the source and the reference translation and fills them in with ChatGPT. The evaluation test: the model passes the test if (i) the original translation quality is good, (ii) the translation quality of the new pair is similar to the old quality. For English and Chinese, the authors show that the new sentences, i.e. the ones used for evaluation, are of good quality. The authors test six MT systems.

The reviewers feel rather ambivalent about the paper. While they acknowledged that the test set could help finding MT errors without additional parallel data, they all had the same concerns. Firstly, it is not clear what to take out of the evaluation results: (i) often the models do not pass the tests because of an error outside of the modified span, (ii) many words/phrases cannot be translated in isolation (e.g., verbs often require collocations with prepositions). Secondly, the core of the method is using ChatGPT to translate: while this might work for the single language pair considered in the paper (English-Chinese), it is largely not applicable for other languages. While the authors replied very extensively and were extremely proactive during discussion, I believe these concerns are (i) valid and (ii) not addressed (e.g., mentioning the use of ChatGPT in the limitation section does not revoke the concern).